# Engineered Zero-Dimensional Fullerene/Carbon Dots-Polymer Based Nanocomposite Membranes for Wastewater Treatment

**DOI:** 10.3390/molecules25214934

**Published:** 2020-10-26

**Authors:** Mona Jani, Jose A. Arcos-Pareja, Ming Ni

**Affiliations:** 1Department of Physical Sciences and Nanotechnology, Yachay Tech University, 100119 Urcuquí, Ecuador; jose.arcos@yachaytech.edu.ec; 2GeneScript, Zhenjiang, Jiangsu 212000, China

**Keywords:** fullerenes, carbon dots, biocompatibility, 0D carbon-polymer nanocomposite membranes, water treatment

## Abstract

With the rapid growth of industrialization, diverse pollutants produced as by-products are emitted to the air-water ecosystem, and toxic contamination of water is one of the most hazardous environmental issues. Various forms of carbon have been used for adsorption, electrochemical, and ion-exchange membrane filtration to separation processes for water treatment. The utilization of carbon materials has gained tremendous attention as they have exceptional properties such as chemical, mechanical, thermal, antibacterial activities, along with reinforcement capability and high thermal stability, that helps to maintain the ecological balance. Recently, engineered nano-carbon incorporated with polymer as a composite membrane has been spotlighted as a new and effective mode for water treatment. In particular, the properties of zero-dimensional (0D) carbon forms (fullerenes and carbon dots) have encouraged researchers to explore them in the field of wastewater treatment through membrane technologies as they are biocompatible, which is the ultimate requirement to ensure the safety of drinking water. Thus, the purpose of this review is to highlight and summarize current advances in the field of water purification/treatment using 0D carbon-polymer-based nanocomposite membranes. Particular emphasis is placed on the development of 0D carbon forms embedded into a variety of polymer membranes and their influence on the improved performance of the resulting membranes. Current challenges and opportunities for future research are discussed.

## 1. Introduction

Hygienic water is vital for the ecological environment and human health. Vast amounts of water deteriorated by contaminants are discharged from industry or through intensification of human activity, thus it is significant to implement conventional water treatments, resource recovery and purification technologies [1]. Increasing demands for advanced water treatments have stimulated an intensive exploration for use of high-performance membrane-based technologies. Membrane-based technologies are exceptionally attractive as they are highly efficient, have low energy consumption, easy scale-up feasibility and have a small carbon footprint [2,3]. Diverse membrane-based technologies have been used for the treatment of water, including micro/ultra/nano-filtration (µF/UF/NF), reverse osmosis and membrane distillation [4]. A common driving force for membrane separation is pressure [5]. Amongst membrane-based technologies, the most common one is commercialized reverse osmosis, which is based on pressure driving forces, consumes high energy and has high operational costs, thus hindering its wider application [6]. For the development of these membrane-based technologies for water purification, membranes made of polymeric materials are attracting increased research interest. Polymeric membranes are energy efficient, can be easily scaled, offer time-saving processes, they are highly permeable to water, have stable structures, are highly water selective, have excellent solute rejection at low operation pressures and are sturdily resistant to oxidation and fouling. Aside from polymer membrane technology, other known processes to purify water are distillation, electrolysis/dialysis, adsorption, chemical oxidation, ion exchange, and biological remediation. For the formation of polymer membranes, polymers such as polyvinylidene difluoride, sulfone polymers, polyacrylonitrile, polyvinyl alcohol/chloride, polyethylene/propylene/ amide, and chitosan are preferred. Some preparation methods for forming polymeric membranes are electrospinning [7], track-etching, stretching, vapor deposition, sol-gel process, phase inversion, and interfacial polymerization (IP) [8]. Thin film composite (TFC) membranes are fabricated using IP, which is essential for commercialization of reverse osmosis and NF processes. Most of these membranes produced via IP have polyamide as a skinny layer on the upper part of a membrane support. The active monomers used to form functional polyamide skinny layers are commonly *m*-phenylenediamine and trimesoyl chloride. The synthetic pathway for preparation of membranes is shown in Scheme 1. The polyamide membranes derived from monomers have good desalination properties [9].

Properties of membranes such as crystallinity, structure, hydrophobicity/hydrophilicity, surface charge and roughness affect their permeate flux, flux rejection, and fouling performances. Most polymer-based water separation membranes are fabricated based on their surface properties that are porous super hydrophobic or hydrophilic [10]. The antifouling properties of hydrophilic membranes are better than those of hydrophobic membranes [11]. Generally, a membrane with higher permeate flux recovery rates exhibits better antifouling properties. The major drawbacks of the available polymer-based membranes are the fouling of membranes caused by the adsorption of surfactants, plugging of pores and structural degradation after long periods in use. The fouling properties depend on the surface characteristics of membranes such as their porosity, hydrophobicity, size and morphology of pores [12]. For hydrophobic membranes, surface roughness and a low surface energy are essential [13], and are achieved by precise surface treatments. With the incorporation of nanomaterials, the surface roughness is increased, and thus polymer nanocomposite membranes are formed. In general, knowledge of nanotechnology comes from the basic elements with certain characteristics. Further, nanotechnology encompasses terms such as nanoscale (about 1–100 nm) and nanomaterials (nano-objects and nanostructured). Nano-objects have dimensions in the nanoscale range, whereas, nanostructured materials have an internal core structure or surface structure that lies in the nanoscale range. When the nanoscale and nanomaterials are jointly present in a polymer/non-polymer matrix they form nanocomposites. Nanocomposites can be defined as nanomaterials which have a multiphase structure which consists of at least one phase of nanoscale dimensions. Nanomaterial properties such as large surface for adsorption, unique surface chemistry, photo- catalysis, antimicrobial, super-paramagnetic, electric and optical properties are beneficial for improving the properties of the resulting material. Nanomaterials could be organic, inorganic compounds or composites. For improving the hydrophilicity and antifouling properties, approaches implemented include IP [14], coating on substrate membranes [15,16], incorporating in situ hydrophilic surface modifying macromolecules [17], grafting [18], blending, or using hydrophilic polymers and monomers [19,20,21], etc. Incorporating various forms of carbon nanofillers to form polymer composite membranes is one of the membrane modification methods. Several nanofillers such as SiO_2_, Al_2_O_3_, Au, zeolites, Fe, Ag, TiO_2_, ZnO, polyhedral oligomeric silsesquioxanes, metal-organic frameworks, etc., are currently used for the formation of nanocomposite polymer membranes. Compared to all these materials, it is critical to know that materials of a carbonaceous nature are environmentally friendly, have low toxicity, facile preparations, they are abundantly available and have excellent biocompatibility. Carbon-based materials are organic compounds containing carbon atoms that exist in the form of spheres, sheets, ellipsoids, nanobuds/ribbons, and hollow tubes. Fullerenes, carbon nanotubes, carbon nanofibers, carbon black, graphene, nanodiamond, CDs, carbon onions, etc. all have different structures and bonding and are the known carbon forms. Carbon nano-forms can be produced in various shapes such as spheres, tubes, sheets, dots and so on. The shape, size and surface properties of nanofillers are important factors affecting the nanocomposite membrane performance, mainly reduced solute selectivity and performance stability.

Carbon-based polymer nanocomposite membranes have recently attracted significant attention for wastewater treatment and purification, mostly for removal of microorganisms, chemical compounds, heavy metals, and separation of oil from water. Figure 1 presents the trend in publications using carbon-polymer nanocomposites developed for wastewater treatment. It can be seen from the figure that the research in the area of wastewater treatment using carbon-polymer nanocomposites is relatively new and the research progress is boosted from 2016 onwards.

With the usage of the carbon nanomaterials it is possible to achieve desirable pore size, larger surface area, and unique surface functionalities that further provides opportunities to enhance the water permeability, thermomechanical stability, improving hydrophilicity, and antifouling properties of polymer-based nanocomposite membranes [22]. All the forms of carbon nanomaterials are biocompatible [23,24,25]. Surface functionalization and interactions of carbon nanomaterials with polymers and approaches to enhance the carbon-polymer interface for the bio-environment have been described in [26] in detail. Antibacterial and photocatalysis characteristics of carbon nanomaterials add advantages to improve the membrane functionality. Consequently, a substantial primary challenge for membrane research lies in improving permeability, rejection, stability and antifouling of carbon-polymer nanocomposite membranes [27,28,29], with a proper characterization of the carbon and carbon-polymer matrix interfaces during loading and assessing the efficiency of load transfer in the nanocomposites.

Looking at the current prerequisites, amongst various carbon forms: 0D fullerene and quantum dots (carbon, graphene, graphene oxide), one-dimensional carbon nanotubes (single-walled and multi-walled), nanofibers and nanohorns, two-dimensional pristine graphene and its derivatives, and ordered mesoporous carbon have been used so far. With large size and higher dimensions as fillers, carbon nanoforms restrict the performance improvements of polymer membranes as it is hard to ensure their real incorporation inside membranes. One way is to meticulously control/reduce the size of nanofillers. With small sized carbon nanofillers the membrane antifouling resistance is enhanced. Further, 0D carbon nanomaterials have proven promising for membrane technology due to their ultra-small size, rich chemical functional groups, and better antifouling properties. 0D carbon nanoparticles result in a smoother membrane surface, smaller pore size, greater permeability enhancement in thin film nanocomposite (TFN) membranes, higher surface hydrophilicity and thus higher water flux composites. Therefore, here we report precisely on 0D nano-forms of carbon materials in polymer membrane composites. Figure 2 is a schematic representation of carbon 0D forms. The crucial aspect of reviewing 0D carbon forms is that they have unique physical as well as surface properties, which help fabricate excellent carbon-incorporated polymer membranes based on a variety of polymers. Above all, the 0D carbon nano-forms are highly biocompatible, which is extremely vital for wastewater treatment, environmental sustainability and human health. In addition, with heaps of research carried out in the field of polymer membrane technology, the 0D carbon nanoforms embedded into polymers to form nanocomposite membranes are rarely reviewed. Therefore, we review the recent insights in the improvement and development of 0D carbon-polymer nanocomposite membranes.

## 2. 0D carbon Nanomaterials (Fullerenes and Carbon Dots): Structure, Properties and Advantages

### 2.1. Fullerenes

Fullerenes are 0D nano-carbons that exist in closed-caged structure with pentagonal and hexagonal rings, represented with the formula of C_20+n_ where n is an integer. Spherical fullerene (C_60_/ buckyball), is widely explored within entire fullerene family. It has the shape of an icosahedron, contains 12 pentagonal and 20 hexagonal rings, a perfect symmetrical cage structure, and is approximately 1 nm in size. Furthermore, carbon in C_60_ has sp^2^ hybridization. C_60_ can resist high pressures (over 3000 atm pressure) and return to the original shape after the pressure is released. C_60_ species have an effective bulk modulus of 668 GPa making it harder than diamond, a high refraction index, a dielectric constant ~4, a large molecular volume, high electron affinities, and large surface-to-volume ratios. C_60_ is traditionally produced by the arc-discharge method, chemical vapor deposition, or by combustion [30]. C_60_ is insoluble or sparingly soluble in several solvents like water. C_60_ dissolve in common solvents at room temperature. Toluene, benzene and carbon disulfide (organic solvents) are the solvents most often used to solubilize them. C_60_ is the only known allotrope of carbon with room temperature solubility, and this allows straightforward processing of C_60_ [31]. C_60_ contains carboxyl, epoxy, and hydroxyl surface groups capable of attracting water molecules. Their adsorption abilities to bond organic molecules through their covalent or non-covalent bonds are good, which facilitates functionalization. C_60_ has a spherical π-conjugated structure and the pyramidalization angle is large due to which chemical functionalization gets easier and higher solubility of C_60_ in many solvents is achieved. Furthermore, interactions of C_60_ with polymers are possible due to their molecular π-electron system which provides minimum transformations. C_60_ can be incorporated into a wide variety of polymers via formation of donor-acceptor or covalent bonds. By inclusion of C_60_ in the polymer matrix_,_ the properties of the polymer changes comprehensively, although the unique properties of C_60_ are retained [32]. Figure 3 shows SEM micrographs illustrating the morphology of poly(phenylene-isophthalamide) membranes modified by 2, 5 and 10 wt% C_60_. The pristine membrane exhibits brittle fracture with a few fracture lines shown as sharp white lines and some plastic deformations shown as rounded white lines in image (a). Fracture surfaces with inclusion of C_60_ (images b-d) contains more plastic deformations, and there is an increase in the density with increasing C_60_ concentration. This shows that the polymer membrane matrix properties are strongly influenced by the carbon nanofiller. C_60_ has been tested by researchers for use in the environment and is found to be beneficial for water treatment. C_60_s have a low aggregation tendency and a high surface area that makes their use as adsorbents in wastewater treatment plausible. C_60_ is preferred due to the low cost of production, easy operation and availability of different adsorbents. C_60_ is ideal as it can adsorb organic compounds in water and is much more effective than soot or activated carbon (a suitable adsorbent, which has a porous structure and large surface area). The ability of C_60_ to adsorb compounds is realized mainly through their interactions in dispersion solutions. Further, C_60_ is hydrophobic, has high electron affinity, large surface to volume ratio, more surface defects, low biological toxicity, and above all it is a photoactive molecule. C_60_ is hydrophobic in behavior and through functionalization it can be turned into a hydrophilic or amphiphilic substance [33,34]. Although beneficial for water treatment applications, the direct use of C_60_ in membrane development is challenging due to its low solubility and poor dispersibility [35]. And in most cases, it must be modified on its surface [8], or forms aggregated (nC_60_) [9], or mixed with a suitable surfactant and stabilizing agents [10].

### 2.2. Carbon Dots

Carbon dots (CDs) are 0D nano-carbon materials with sizes lower than 10 nm, displaying bright fluorescence (the highest quantum yield over 90%), with low toxicity and superior photostability [36]. CDs have intrinsic emission derived from the quantum confinement effect due to the existence of multiple photoluminescence centers. The observed photoluminescence is size independent, excitation dependent, has a broad emissive full width at half maxima and short lifetimes. CDs have superior optical properties such as strong absorption, bright photoluminescence, excellent light stability (resistance to light decomposition, photobleaching and blinking). Most CDs contain sp^2^-π bonds, typically alike to nano crystalline graphite, although without structural identification [37,38,39,40,41]. Concretely they are classified as carbon nanodots (CnDs), carbon quantum dots (CQDs), graphene/graphene oxide quantum dots (GQDs/GoQDs), and carbonized polymer dots (CPDs). All of these types have similar sizes and photoelectrochemical properties, although they differ in carbon core structure, and surface chemical groups [42]. Amorphous quasi-spherical nanodots that lack quantum confinement are considered to be CnDs. CnDs are mainly prepared by pyrolysis processes or by polymerization using tiny molecular precursors [36,43,44]. While spherical quantum dots with quantum confinement and crystalline structures are referred as CQDs, the optoelectronic properties of these quantum dots can be altered and enriched upon surface passivation or functionalization. The π-conjugated single graphene sheets are referred to as GQDs. By pyrolysis methods with graphite as starting material it is possible to exfoliate graphite into a few-layers of GQDs. GQDs must not be mistaken with CDs. The of core CDs is mainly composed of sp^3^-hybridized carbon, usually amorphous and spherical, with less than 10 nm in size. On the other hand, GQDs are a disk of graphene in the 2–20 nm size range and are composed mainly of sp^2^-hybridized carbon. They are crystalline and have “molecule-like” character rather than colloidal. Quantum confinement is still not well understood for GQDs. GQDs can have different sizes and the same bandgap energy (i.e., 3.4 eV), however, the bandgap energy of pristine graphene is 0 eV or close to 0 eV, that’s why they are named quantum dots when compared to graphene. Although GQDs have identical photoluminescence and similar emission properties to CDs. GQDs have molecule-like character and thus show tunable optoelectronic properties. The position of the absorption peak is not influenced by the size of GQDs, unlike other CDs. CDs and GQDs have complex surface functional groups, especially oxygen-related functional groups, such as carboxyl and hydroxyl. The surface groups contribute greatly to the optical properties of CDs and GQDs and even make them water-dispersible. CPDs have a mixed polymer-carbon structure and a carbon core. They comprise abundant functional groups and polymer chains on their surface. CPDs are obtained due to the incomplete carbonization of the polymer clusters while using the hydrothermal or solvothermal processes [45]. CPDs have prominent optical properties similar to CDs and GQDs, and in addition have the privilege of polymer properties such as abundant functional groups, short polymer chains and highly crosslinked network (polymer/carbon hybrid) structures. CPDs have excellent aqueous solubility and outstanding photoluminescence quantum yield as compared to CDs and GQDs. CPDs mostly possess strong blue and green emission, even up to several long wavelengths. They have strong absorption in the UV region and have excitation dependent emission. CPDs with dual-emission fluorescence intensities (red and blue emission) prepared in different pH conditions, were used to distinguish between four types of bacteria [46]. CPDs are newly emerging luminescent CDs and still lack use in membrane-based water treatment research.

CDs are mostly prepared using physical or chemical methods, denoted as “top-down” and “bottom-up” methods. With the development of microwave and hydrothermal technologies, various “bottom-up” methods were explored to prepare CDs from small molecules, graphite, polymers, biomolecules, and biomass [47,48,49]. Figure 4 is a schematic diagram which shows the formation of CDs from glycine and the formation includes dehydration, polymerization, carbonization and passivation steps [50]. CDs possess many functional groups on their surface which includes amines, epoxy, ethers, carbonyls, hydroxyls, and carboxylic acids [51]. Plenty of functional groups on the surface of CDs make them highly hydrophilic in nature and provide opportunities to functionalize them with a variety of organic, polymeric, inorganic, or biological species [26,52,53,54,55]. Since CDs are hydrophilic, they show good solubility and stability in water. CDs exhibit excellent biocompatibility, which can allow exposed cells or organisms to live sustainably, even at high concentrations [56,57,58]. In addition, CDs form high performance NF membranes and break the trade-off effect between water permeability and selectivity. Besides, GQDs have a strong sorption capacity for heavy metals compared to other substrates. The large amount of surface groups/polymer chains, such as carboxyl, hydroxyl, amine, etc., give rise to their excellent water solubility and convenience for forming composites with other materials without phase separation. TFNs with improved water permeability, antifouling performance, bactericidal effects or mono-/divalent ion separation capacity have been successfully developed. GoQDs-based composite membranes have favorable water permeability and are anticipated to be a creative filler to capture water molecules and provide shorter diffusion pathways in the membranes. Currently, the green synthesis (i.e., where the starting materials are non-toxic reagents, eco-friendly and biosafe) approach has gained a lot of popularity in the field of CQDs synthesis which has several advantages for the environment [57].

## 3. Biocompatibility Implications of 0D Nano-Carbon in Water Effluent Treatments

During the process of wastewater treatment using carbon-polymer nanocomposite membranes, it is possible that the 0D nano-carbons might enter and remain in freshwater ecosystems. Thus, this section describes the study of their biocompatible impact. C_60_ is apt as a representative material for environmental studies. C_60_ is reported to be either not cytotoxic or harmful under specific conditions [59,60], and have biological consequences that are neutral [61,62,63,64]. The antibacterial activities of water-soluble C_60_ derivatives [65,66,67] or nC_60_ have been investigated, and the studies have shown that when C_60_ were prepared under specifically low salt conditions, they were found to be toxic to bacteria [68,69,70]. An anaerobic biodegradation of wastewater sludge was performed using C_60_. The analysis showed no significant effect on the structure or function of the anaerobic community [71]. For a basal soil respiration study, 1 µg of C_60_ per gram of soil in aqueous suspension or 1000 µg C_60_ per gram of soil in granular form were used. From denaturing gradient gel electrophoresis profiles a slight impact on the structure (as shown in Figure 5) and on the function of the soil microbial community and processes was seen [72]. Furthermore, C_60_ is not cytotoxic towards human and animal cells in vitro although acute toxicity is observed in animal tissues in vivo [73,74,75,76,77,78]. By modifying the surface, the C_60_ can interact differently to the biological molecules and make them cytotoxic [79,80]. Thus, likely only modified C_60_ could be cytotoxic. C_60_s are used as bio-receptors and sensors, as they are biocompatible with living organism-based nanomaterials [81,82,83,84,85]. Overall, C_60_ is non-toxic and can be safely exploited for water treatments such as filtration, as adsorbents, and membrane technologies for the environment.

Few studies have examined the environmental behavior and toxicology of CDs on natural mineral particles [86,87,88]. Individual types of CDs possess distinct physicochemical properties, which in turn determine their potential toxicity. Generally, the carbon itself is not toxic and if any cytotoxicity is reported it is primarily due to the surface passivating agents used [89]. Even if CDs are modified with highly cytotoxic profile agents, they still can be used for in vivo applications, provided specific conditions are used such as low concentrations and short incubation times. CDs are safe for health and free of environmental concerns [89,90] and have been developed for nanobioprobes and clinical treatments. CDs (with or without surface passivation) have low toxicity and can be internalized into cells for imaging purposes [91,92]. CDs have been demonstrated to cause no cytotoxicity at concentrations of approximately 0.1–10 µg/mL (which is 10–1000 times higher than the normal amount required for imaging applications) [40]. The synthesis of CDs using green synthesis methods represents an improvement in biocompatibility and low cytotoxicity, which is crucial for the environment [57].

## 4. Development and Influences of 0D Carbon Nano-Forms Incorporated in Various Polymers to Form Nanocomposite Membranes

### 4.1. Fullerene-Incorporated Polymer Nanocomposites

C_60_ is often chemically functionalized so as to enhance the mixing capability of C_60_ with other host polymers [93,94]. A sequence of studies have shown that when the polymer membranes are modified with C_60_, the membranes’ initial properties are improved [95]. The outstanding activity towards the damage by radicals, and excellent thermal as well as antifouling performance are reported. Usage of aromatic polyamides, such as polyphenylene isophthalamide shows promise for NF, UF, distillation, and reverse osmosis. These polymers are mechanically resistant, chemically stable, have low cost, ease of workability and rigidity, and have high porosity [96,97,98,99]. Novel membranes are made using aromatic polyamides, typically modified by polyhydroxylated C_60_ (fullerenol/C_60_(OH)_n_), carboxy C_60_ and C_60_ derivatives with L-arginine via a solid phase synthesis route. These C_60_-modified polyamide membranes show high permeation fluxes and enhanced selectivity [100]. With the modification of aromatic polyamides by C_60_ derivatives, the structural environment of the polyamide changes (due to the noncovalent bonding between them) and the internal composite membrane structure changes. In addition, it is observed that the surface hydrophilicity, membrane density and surface roughness increases. Figure 6 shows the graphical form of mixed matrix pervaporation (PV) membranes prepared by Dimitrenko et al., and Table 1 shows the PV separation indexes. From Table 1 it can be seen that aromatic polyamide membranes modified with fullerenol shows the best transport properties for the PV of azeotropic methanol-toluene (72/28 wt%) mixtures. In addition, in comparison to other C_60_ derivatives, membranes containing fullerenol show the highest permeation flux (0.649 kg/(m^2^h)), and enhanced selectivity with respect to methanol. In another report, a polyphenylene isophthalamide membrane was modified by adding 10 wt% C_60_ via a solid-phase method to form nano-UF membranes. With the increase in C_60_ content the membrane rigidity was enhanced and showed improvement in its technological parameters [96]. In addition, polyphenylene isophthalamide with 10 wt% C_60_ membranes showed increased flux, reduced recovery (0.8–0.9) and lower protein sorption [101]. Polyphenylene isophthalamide with C_60_ improves the PV properties as well [32]. Furthermore, the physical properties of membranes such as the intrinsic viscosity are significantly influenced by the inclusion of C_60_ in a polyphenylene isophthalamide matrix. With the inclusion of C_60_, the structure of the membranes becomes more compact, denser, and reveals a non-monotonic effect on the glass transition temperature.

Asymmetric polymer membranes based on the hydrophobic polymer poly(2,6-dimethyl-1,4-phenylene oxide) (PPO) with inclusion of 2 and 10 wt% C_60_ are prepared using a solid-phase interaction method. These PPO-C_60_ membranes are prepared to study removal and adsorption behaviors of estrogenic compounds. The morphology of the membranes showed an increase in the pore size and porosity on the dense top layer of PPO-C_60_ membranes as compared to pure PPO membrane. The permeation flux is reported to be higher for PPO-10 wt% C_60_ membranes. Notably, the effect on the inclusion of C_60_ in PPO matrix depends on the approach used for modifying the matrix [32]. Changes in the polymer properties after modification by C_60_ have been demonstrated using polymers such as poly(vinylpyrrolidone) (PVP) and polystyrene (PS) [102,103,104]. By spectroscopic analysis, it has been proven that donor-acceptor interactions exist between the polymers and C_60_ in PS-C_60_ [104] and PVP-C_60_ [96] complexes. It is reported that membranes formed using fullerenol and PVP-C_60_ are useful in water treatments as these membranes can help to target specific pollutants or microorganisms in the water, and are more sensitive to superoxide or singlet oxygen [105]. C_60_ and C_60_(OH)_12_ -doped Nafion composite membranes were fabricated by Tasaki et al. through a solution casting method. This method opened the possibility to directly incorporate C_60_ into immiscible polymers without any chemical modification. By using this method, the characteristics of C_60_ and C_60_(OH)_12_ were retained in the Nafion composite membranes [106]. Photoconductivity and antimicrobial activity studies have been performed using C_60_ incorporated into a variety of polymers such as polycarbonate, polyethylene, PS-polyisoprene-PS, PS-polybutadiene-PS, polythiophenes, poly(bromostyrene), poly(*n*-vinylcarbazole), and 1,4-polydiene [107,108,109,110,111,112,113].

### 4.2. Carbon Dot-Incorporated Polymer Nanocomposites

CDs have plenty of hydrophilic carbonyl and carboxyl groups on their surface which benefits their uniform dispersion in water. In addition, these surface functional groups provide an immense tendency to get attached to the pendant polar groups that are present in polymers. Further, these functional groups help to ease the membrane fabrication process while incorporating CDs into polymers and provide better membrane performances [114,115,116,117]. The working span of polymer composite membranes gets affected due to stress dissipation, lack of reinforcement homogeneity and may suffer from thermal stability. In this context, CQDs act as distinct reinforcers by providing uniform-dispersion, selective transport sites for separation membranes and play a significant role in the remediation. Few other advantages of CDs are their carbon core, finite tuned size, good dispersion in organic/aqueous solvents [116] and ease of synthesis [118]. Overall, their nontoxic nature permits a high potential use of CDs for modifying polymer membrane properties for water treatment uses. For the fabrication of polymer nanocomposite membranes, the surface chemistry of CDs is tuned so as to accomplish better membrane stability and performance as reported in references [114,115,119]. Sun et al., tuned the CQDs with a variety of functional groups such as carboxyl, amino and sulfonic acids and incorporated them in a polyamide layer via IP and studied the properties of the resulting membranes. The membranes functionalized with sulfonic acid functional groups on CQDs were reported to have a permeate flux of 42.1 L/m^2^h and a Na_2_SO_4_ rejection of 93.6%, and was endowed with the best antifouling performance. These changes in the membrane properties were due to the formation of looser polyamide chemical structures and a largely negatively charged membrane surface due to the incorporated CQDs, whereas, the membranes functionalized with amino-modified surface functional groups on CQDs exhibited better retention properties and exhibited a less negatively charged membrane surface compared to the non-functionalized CQDs-polyamide composite membranes [117]. CD-based membranes for water treatment are formed by two main approaches. First is CDs incorporated into a thin polymer layer, known as TFN membranes, where typically CDs are dispersed in the aqueous phase and subsequently contribute in the IP process to form TFN membranes. TFN membranes are produced via techniques such as coating in addition to IP processes, although large-scale preparation is a challenge. Secondly, mixed matrix membranes are composed by adding CDs in polymer matrices to form homogenous solutions via various spinning methods. Here the challenges are the uniform dispersion of CDs and leaching [120]. Approaches such as coating of CDs on the top layer of membrane surface are also reported.

Polyamide TFN membrane made by IP techniques differ in characteristics (higher water flux, separation capability, pH tolerance) from the asymmetric membranes which are formed via phase inversion techniques. Here, it is worth mentioning that particles with larger size allow faster fouling of membranes [121]. Conversely, using ultra-small sized nanoparticles, such as CQDs allow one to significantly enhance the membrane antifouling resistance [122,123]. Li et al. incorporated CDs which are super hydrophilic and have quantum sizes of 6.8 nm into a polyamide layer. The incorporation of such CDs led to higher surface hydrophilicity and water flux for the formed membranes [116]. Bi et al., fabricated TFN membranes incorporated with ~2 nm GQDs via IP of piperazine and trimesoyl chloride. The GQDs were added as aqueous additives into the membranes and poly(ether sulfone) was used as support membrane. Addition of small-sized GQDs efficiently tuned the surface roughness, membrane structure and hydrophilicity of the formed TFN membranes [122]. Figure 7 is a schematic representation of flow of water passing through the membrane channels at the interface between the GQDs and the polyamide layer. These membranes present excellent water permeation, due to a synergistic effect of the surface hydrophilic GQDs. Additionally the graph in Figure 7 shows the antifouling properties of the GQD-based polyamide membranes that are assessed through a dead-end filtration experiment. The foulants used were bovine serum albumin, humic acid and emulsified oil. The results showed that steady water fluxes under harsh fouling conditions could be achieved using GQD-polyamide TFN membranes.

Another report showed that nitrogen-doped GoQDs-polyamide TFN membranes could be developed. The formed amine groups on the GoQDs surface due to nitrogen doping were used as linkers to form chemical bonds between GoQDs and the polyamide matrix. The GoQDs formed stable dispersions, with improved thermal stability and surface hydrophilicity. Further, the water permeability increases thrice with maintained salt rejection, which is promising for high flux water desalination applications [119]. µF and UF membranes made from poly(vinylidene fluoride) (PVF), are widely used in industrial wastewater treatment, as the PVF membranes have superior chemical and thermal stability, high resistance to radiation and strong mechanical properties [124]. Zeng et al. used covalent bonding of GoQDs onto amino-modified PVF membranes, and found improved hydrophilicity, anti-bacterial, anti-fouling performance and an increase in water flux [125]. Moreover, the water contact angle was reported to decrease from 118.5° to 34.3° due to the coating of GoQDs on the PVF membrane surface. Novel GQDs-PVF nano-fibrous mixed matrix membranes are prepared for water desalination via an air gap membrane distillation process [126]. By adding GQDs the formed PVF membrane structure is more compact, has rougher surface and higher wetting resistance. CDs with tailored functional groups were facilely synthesized and embedded into polyethyleneimine matrix, and then dip-coated on polyacrylonitrile support to prepare composite membranes. The method used is IP, and these membranes are prepared to study polar organic solvent transport across the membranes for NF. The low and high carbonation degrees of CDs were obtained by decreasing glycerol mass in the reaction solution while modifying the surface of CDs. The low carbonation CDs could facilitate polar solvent migration through the membrane by providing bonding sites of hydrophilic groups (-OH, -CO_2_H, -NH_2_). In contrast, high carbonation CDs showed an increase of non-polar solvent uptake and permeation via their hydrophobic domains [114]. Another report shows that the membrane hydrophilicity is improved by Na^+^ functionalization on CQDs [127,128], as the existence of Na^+^ containing groups facilitates uniform dispersion of CQDs in aqueous solutions. Moreover, Na^+^-containing CQDs exhibited the highest water flux of 53.54 L/m^2^h and power density of 34.20 W/m^2^ for pressure retarded osmosis membranes [115]. The Na^+^ functionalized on CQDs is dispersed during the IP in the polyamide selective layer to form novel TFC membranes, and the membranes showed effective changes in the surface structure of membranes due to their rich functionality and small size. The membranes were prepared to remove heavy metals via NF [128]. TFN membranes incorporated with GQDs embedded in a polyamide matrix via IP of piperazine and trimesoyl chloride were fabricated. The formed GQDs-polyamide TFN membranes exhibited enhanced water permeability and antifouling properties [122]. The amine groups of piperazine and hydroxyl or carboxyl groups of GQDs reacted with the acyl chloride groups of trimesoyl chloride at the oil/water phase zone during the IP method, which resulted in an ultrathin polyamide layer on the porous substrate. Nanocomposites consisting of CDs and polypyrrole, with high electrical conductivity exhibited high selectivity and sensitivity for the detection of trace amounts of picric acid that are present in water and soil [129]. Polyacrylonitrile-CQD composite nanofibers were produced by electrospinning and were characterized. And these composites could have possible future applications in wide areas of research such as smart clothing, high-performance aircrafts, sensors, photochemical reactions, biological imaging, and optoelectronic devices [130]. Most of the prepared polymer membranes based on 0D carbon forms are spherical dot-like, agglomerated or sheets and are in the form of TFC (with nanocomposite substrate) and TFN (with porous substrates or surface-coated TFC). Few researchers have prepared TFC on hollow fiber membranes. Efficient polymer membranes are prepared using nanofibrous GQDs. Reports show that by incorporating CDs the membranes form finger-like structural morphology with a smooth surface, and even porous fingerlike macrovoid structural membranes are reported using GoQDs sheets. C_18_-CQDs that have a knitted structure were used to prepare TFN membranes with a substrate composed of fibers. C_60_(OH)_24–28_ with TFN showed a membrane surface with leaf-like structures. Precisely, incorporation of nano-sized CDs, irrespective of their shape, into the polymeric matrix to form membrane enhances the membrane properties which are mostly the mechanical strength and antifouling property.

## 5. 0D Carbon-Incorporated Polymer Nanocomposite Membranes for Wastewater Treatment

Carbon-polymer nanocomposite membranes are prepared by incorporating various forms of nano-carbon (filler) into a polymer matrix [131,132]. The carbon-polymer nanocomposite membranes are used in broad application areas for desalination, antibacterial applications, and removing inorganic contaminants, dyes, natural organic matter, separation of nano-matter, water flux oil rejection, and emerging contaminants of concern. Mainly, these 0D carbon nano-forms incorporated into polymer matrices have demonstrated immense capability and potential for eliminating various water pollutants such as pathogens, heavy-metal ions, and recalcitrant organic compounds [133,134,135]. Thereby, desired efficient water-treatment technologies such as 0D carbon-polymer nanocomposite membranes can act as supplements or substitutes for the traditional ones in the future.

### 5.1. Fullerene-Based Polymer-Nanocomposite Membranes

In studies by Brunet et al. hydrophilic functionalized C_60_ species were prepared and by utilizing the photocatalytic property of C_60_ they could be used to kill pathogenic microorganisms that are present in water [105], thus showing the benefits of C_60_ for water treatment. Further, sorption is one of the methods to get rid of heavy metals such as cadmium, lead, zinc, nickel, cobalt, copper, arsenic, and mercury, etc., from wastewater effluents. The sorption capacity of metals is usually associated with surface defects and the lattice structure of the material used [136,137]. Conventional materials have low metal sorption capacity and low metal removal efficiency while treating wastewater. By using C_60,_ a porous structure was developed with an increase in the hydrophobicity of the prepared adsorbents and the results showed an improvement in metal sorption capacity. For instance, using 0.001–0.004% of the C_60_ in activated carbon, the sorption capacity for heavy metals such as lead (II) and copper (II) increased by 1.5–2.5 times [138]. Additionally the electronic properties of C_60_-based composite materials could be utilized as they have been reported to show higher specific capacitance of 135.36 Fg^−1^, and better retention time [139]. Alekseeva et al. reported that a C_60_-based nanocomposite-PS film which had better efficiency for the removal of Cu^2+^ ions, following a Langmuir model [140]. The fabrication of C_60_-based polymer film increases its hydrophobicity, which makes them better in adsorption and easier in recycling [141]. Asymmetric UF membranes based on poly(phenylene isophtalamide)-C_60_ composite membranes were prepared by a phase inversion technique [96]. The phosphate buffer flux reduced recovery was estimated by static sorption tests. For the static sorption tests, the membranes were immersed in a protein solution for 20 h. The results showed that with an increase in C_60_ content, the protein adsorption decreases on the membrane surface and shows better values of flux reduced recovery rates. Dmitrenko et al., studied the transport properties of dense polyphenylene isophthalamide membranes modified by C_60_ and its derivatives, and tested the PV separation of methanol/toluene mixtures, including azeotropic compositions [100]. The results showed an improved permeation flux of 0.084–0.214 kg/(m^2^h) and a high level of selectivity. Antibacterial membranes prepared by grafting C_60_ with PVP showing the safety of using C_60_ have been reported for water disinfection. C_60_s act as nano-adsorbents in the membranes and improve the membranes’ adsorption efficiency. Hydrophobic PPO membranes incorporated with various compositions of C_60_ were studied for removal and adsorption behaviors of the natural hormone estrone. The results showed the importance of membrane pore size and internal structure [142]. C_60_ incorporation in hydrophobic polymers improved 8-fold the permeate flux compared to pristine polymer membranes. For long term filtration, 10 wt% C_60_-PPO nanocomposite membranes showed good removal performance of at least 95% of permeate, attributed to C_60_′s adsorption capabilities and steric hindrance effects. Plisko et al. fabricated novel polyamide-C_60_(OH)_22–24_ TFN hollow fiber membranes [143]. C_60_(OH)_22–24_ was incorporated via an aqueous phase in triethylenetetramine onto the polysulfone substrate during IP. The TFN membrane containing 0.5 wt% of C_60_(OH)_22–24_ demonstrated the best antifouling performance for removal of the organic matter. Perera et al. fabricated C_60_(OH)_24–28_ incorporated TFC membranes for forward osmosis by IP process, showing improved specific desalination performances [144]. Superior desalination performances such as water flux, reverse salt flux, antifouling propensity, water permeability and salt permeability of the fabricated C_60_(OH)_24–28_ based TFN membranes were presented. Introduction of various C_60_(OH)_24–28_ loading on the polyamide topmost surface yielded an increase of pure water flux, decreased salt rejection, and superior antifouling performance. With a loading of 400 ppm C_60_(OH)_24–28_, a water flux of 26.1 L/m^2^h, higher than that of the pure TFC membrane was reported. Shen et al. developed a novel TFN membrane by loading fullerenol via IP. With 0.01% (w/v) fullerenol, the membrane revealed excellent antifouling ability, stable and high efficiency in Mg^2+^/Li^+^ separation with a high separation factor of 13.1. These membranes formed were suggested to have great potential in the recovery of Li^+^ from seawater [145]. Liu et al. reported C_60_ grafted graphene oxide membranes with a fixed interlayer spacing around ∼12.5 Å [146]. Figure 8 shows the fabrication process, the water desalination setup and the schematic representation of blockage of anions and cations through the membrane. The membranes were reported to obtain a high water flux up to 10.85 L/m^2^hbar (which is high enough for brackish water desalination), and 0.1883 mol/m^2^hbar ion permeation rate at an applied pressure of 5 bar. Although C_60_ has great potential for water adsorption application, the cost of production into large quantities is high, which restricts their convenience in utilization. Thus, there are very few reports on C_60_ for wastewater treatments so far.

### 5.2. Carbon Dot-Based Polymer-Nanocomposite Membranes

CDs can be used as adsorbents to remove contaminants from wastewater [147]. Wang et al., reported the formation of periodic mesoporous organosilica embedded with CDs and adopted them as an adsorbent for the removal of toxic organic pollutants (2, 4-dichlorophenol) and inorganic metal ions (mercury (II), copper (II) and lead (II)). The adsorptions followed Langmuir and Freundlich models and obeyed pseudo-second-order kinetics [147]. CQDs have high-performance efficiency in water treatment membranes as they are hydrophilic by excellence, have desirable size, tunable surface functional properties, and favorable polymer affinity. It is shown that the separation performance of the CQD-based polymer membranes can be effectively modulated by tuning the functional groups on the surface of CQDs [117]. A 5 nm CQD with tunable functional surface groups i.e., low and high carbonation degree, was easily synthesized and embedded into a polyethyleneimine matrix, and then dip-coated on a polyacrylonitrile support to prepare composite membranes. The prepared CQD-based NF membranes were fabricated for separation of organic solvents. Solvent resistance, solvent flux, and solute rejection were evaluated. It was observed that low carbonated CQD slightly suppresses the uptake and permeation for non-polar solvents. Conversely it enhances permeation for polar solvents. The membranes which were prepared with highly carbonated CQD acting as a non-polar solvent accelerator through their hydrophobic domains, and shows that the permeation of polar solvents is blocked [114]. CQD-based NF membranes have been proven efficient for biogas slurry valorization to reduce the environmental pollution [148]. The membranes prepared for biogas slurry valorization consisted of hydrophilic CQDs interlayered between the substrate and selective TFC NF membrane layers. CQDs as interlayers resulted in an enhanced water permeation of the NF membranes as they provided channels for fast water and ion transport, thus demonstrating a fantastic separation performance. CQDs as excellent membrane modifier for the desalination and wastewater treatment have been reported by Koulivand et al. [149]. For modifying the membrane properties, CQDs were synthesized by a pyrolysis method and were added to the polyether sulfone casting solution using a non-solvent induced phase inversion technique. Addition of CQD into the polyether sulfone membrane matrix, resulted in improved membrane morphology, porosity, surface charge, permeability (76.5 kg/m^2^h), and enhanced fouling resistance of the membrane. The fouling resistance was enhanced due to the decreased water contact angle and increasing surface hydrophilicity provided by the incorporation of CQDs. In addition, separation tests of reactive red 198 dye and salts (Na_2_SO_4_, MgSO_4_, and NaCl) showed higher retention performance due to the presence of CQDs in membranes. Super hydrophilic 6.8 nm CQDs (0.02 wt%) were incorporated into the selective layer of polyamide TFN reverse osmosis membrane [116]. The CQD-polyamide TFN membrane exhibited promising desalination performance with a water flux of 87.1 L/m^2^h, salt rejection of 98.8% for long durations. Zhao and co-workers reported 3.2 nm-sized CQDs immobilized onto the polydopamine layer which is grafted on the surface of poly(ether sulfone) substrate were prepared for pressure retarded osmotic power generation and waste water treatments. Due to the immobilization of CQDs, the membranes possessed high power density, enhanced antibacterial and anti-biofouling activity [32]. Na^+^ functionalized CQDs have been preferred in the forward osmosis membranes as it is reported that the presence of CQDs draw solutes and attain the highest water flux for seawater desalination [127]. A comparative study was carried by Gai et al. where three kinds of CQDs (i.e., original, Na^+^ functionalized at pH 5 and pH 9) were synthesized and then embedded into polyamide layers as pressure-retarded osmosis membranes [115]. It was demonstrated that the Na^+^ functionalized CQDs with pH 9 exhibited the highest water flux of 53.54 L/m^2^h and a power density of 34.20 W/m^2^. Na^+^ functionalized CQD nanofillers were used to develop TFN hollow fiber membranes via IP for brackish water desalination. Water and salt permeability, water flux and solute rejection using the prepared membranes were carried out. The water flux and salt rejection were increased to 53.54 L/m^2^h and 98.6% with the loading of Na^+^ functionalized CQDs as nanofillers. Although precise measures to design of a polymer-CQD nanocomposite are lacking, it may reveal a huge potential in water treatments [150]. Moreover, He et al. reported the influence of Na^+^ modified on CQDs for the formation of polyamide TFN membranes. The surface structure and hydrophilicity of the formed TFN membrane were improved due to the uniform dispersion of CQDs possible due to Na^+^ functionalization [128]. The influence of Na^+^ modified on CQDs (0.05 wt%) showed water permeability of 10.4 L/m^2^hbar and impressive rejections of 97.5%, 98.2%, and 99.5% towards SeO_3_^2-^, SeO_4_^2-^ and HAsO_4_^2-^, with a superior antifouling property and robust long-term stability. The formed membranes due to functionalization of Na^+^ are anticipated to show improved separation performances of selenium and arsenate ion contaminations in surface and ground water. Lei et al. functionalized superior hydrophobic C_18_-CQDs by grafting CQDs with octadecylamine [151]. Superior hydrophobicity was achieved by cross-linking C_18_-CQDs with cotton textile using tolylene-2,4-diisocyanate. These hydrophobic C_18_-CQDs membranes were prepared for the separation of oil-water (99%) and exhibited unique selectivity, feasible for water desalination. Owing to the photocatalytic properties of CQDs, NF CQDs-polydopamine membranes are fabricated as durable self-cleaning membranes (the fabrication process is shown in Figure 9) [152]. The insertion of the polydopamine–photoactive CQDs sandwich can degrade organic molecules adsorbed on the surface of the membrane under visible light, and show it is promising for low-cost fouling remediation and for self-cleaning.

Punrat et al. prepared polyaniline-GQD nanocomposite membranes to assess Cr (VI) levels in mineral drinking water and in deteriorated Cr-plating samples. The recovery rate was 80.3–106% [153]. Bi et al. [122] prepared GQDs-polyamide TFN-NF membranes which are reported to have a maximal water permeance of 510 L/m^2^hMPa. This water permeance is nearly 6.8 times higher than that of the pristine polyamide membrane and it has good antifouling performance. In another report, Bi et al. [154] reported a GQDs incorporated in NF membranes that exhibited an ultrafast water permeance of 244.7 L/m^2^hbar, about 5–6 times higher compared to previous reports, with a rejection of 92.9% and 98.8% for Alcian blue and Congo red. Wu et al. fabricated a GQDs-based solvent resistant NF membrane via IP on hydrolyzed polyacrylonitrile support and the acetonitrile and hexane permeances reached 469 and 508 L/m/^2^hMPa, respectively [155]. Li et al. [156] prepared GQDs-polyimide TFN membranes with improved solvent resistance and achieved higher ethanol permeances. The membranes had a sandwich-like structure using low concentrations of m-phenylenediamine and trimesoyl chloride during IP. Due to the incorporation of GQDs the membrane thickness (about 25 nm) was reduced and exhibited ultra-low surface roughness (average less than 2 nm) Further, the GQDs incorporated membranes showed an increased Rhodamine B rejection (from 87.4% to 98.7%) and an increased ethanol permeance (from 33.5 to 40.3 L/m^2^h MPa ^− 1^). In addition, the prepared membranes have superior solvent resistance, antifouling properties for long durations. GQDs functionalized with amino groups to form TFN membranes for solvent resistant NF membranes are reported [157]. The membranes exhibited excellent solvent resistance in strong polar solvents at high temperatures. Seyedpour et al. [158] incorporated nano-sized bactericidal GQDs in the active layer of forward osmosis membranes. The membranes’ antimicrobial activity was improved and better forward osmosis performance was achieved. Xu et al., reported GQD-polyethyleneimine TFC membranes for forward osmosis desalination [159]. The covalent bonds formed between GQDs and polyethyleneimine helped to improve the stability of the membranes during filtration and hydraulic cleaning processes. The membrane loaded with 0.050 wt% GQDs had a hydrophilic and neutrally charged membrane surface, exhibiting enhanced water flux of 12.9 L/m^2^h, and good anti-fouling performances. Thus, GQDs-based polymer nanocomposite membranes present great potential in applications such as desalination, purification and wastewater treatments.

Further, GoQDs have a particular size, shape, and edge structure and an excellent dispersion into the polymer matrix, which is desirable for separation and permeation applications, although few reports exist for water purification membranes using GoQDs. Song et al. were the first to incorporate GoQDs as nanofillers to form TFN reverse osmosis membranes. The membranes showed improved antifouling and chlorine resistance for desalination and water reclamation applications [160]. Fathizadeh et al. [119] fabricated nitrogen-doped GoQDs polyamide TFN membranes via IP. The membranes’ water permeance was 2-fold more than that of the membranes without GoQDs with 93% salt rejection. Zeng et al. reported covalently bonded GoQDs onto amino-modified polyvinylidene fluoride TFN membranes. Due to the unique structure and uniform dispersion of GoQDs in the membranes, the UF membranes had enhanced bactericidal, anti-biofouling performances, long-term stability and durability [125]. The water flux permeation increased from around 500 L/m^2^hbar to >3800 L/m^2^hbar. The membrane’s hydrophilicity was improved with a decrease in water contact angle. Zhang et al., fabricated a low-pressure GoQDs based tannic acid film TFN NF membrane by IP [161]. The Congo red and methylene blue rejections were 99.8% and 97.6%, with a water flux of 23.33 L/m^2^h, due to the improved hydrophilicity, smooth, and negatively charged surface of the formed membranes. Nitrogen-doped GoQDs (0.02 wt%)-based polyamide TFN membranes were prepared which showed a drastic change in the water flux, with preserved high salt rejection due to a superior thermal stability, improved hydrophilicity, and a higher effective surface area [119]. GoQDs have been integrated into poly(vinyl alcohol) and evenly cast on a polysulfone support membrane for PV. By integrating GOQDs on the membrane, the separation performance was changed with excellent dehydration of alcohol/water mixtures [162].

In Table 2, the average quantum size of all the 0D carbon nanofillers mentioned above used in polymer membrane technology is in the range between 1 nm to 20 nm. When the nanofillers are functionalized or surface-modified, the size of the 0D carbon forms increases. The resulting membrane pore size or thickness changes and is dependent on the concentration of the nanofillers. For a particular case, blockage of the pores due to the presence of nanocarbons is also mentioned. From Table 2, a comparative study shows that the permeation flux is higher when using GQDs and GoQDs as nanofillers in polymer nanocomposite membranes and the solvent rejection is on average mostly above 90% for all kinds of nanocomposite 0D carbon fillers.

Furthermore, the 0D carbon nano-forms are preferred over other types of fillers as they are ecofriendly, have easy fabrication processes, reduce the environmental pollution and enhance the economic profit. From the above mentioned 0D carbon fillers for polymer membrane fabrication, the resulting membranes opt to show good antifouling and anti-biofouling properties, more compact and stable structures, durability for long operating times (even at higher temperatures), superior resistance to chemical reagents, desirable surface hydrophilicity/hydrophobicity, enhanced membrane density, low toxicity, and long-term organic solvent stability. These membranes are mostly fabricated for separation (biomolecules, oil-saltwater, selenium and arsenic, metals), self-cleaning, desalination, purification and wastewater treatments.

## 6. Conclusions and Future Outlook

The review presents the fabrication of 0D carbon nanomaterials such as C_60_ and CDs used as nanofillers to incorporate them into polymers to form nanocomposite membranes. These carbon nanomaterials possess extraordinary properties, biocompatibility, and ease of fabrication, that have proved to be a leap forward in opportunities to revolutionize their potential for desalination and separation processes for wastewater purification. The astonishing performance of C_60_ or CDs -incorporated into a variety or polymers to form nanocomposite membranes and several approaches adopted to improve the membrane performances is revealed herewith. Numerous efforts have been focused on improving the nanofiller-polymer membrane properties (particularly their water permeability, separation efficiency, and antifouling performances), searching how to efficiently blend these nanofillers into polymers, surface modifications, cost-effectiveness and their long-term stability. It is established that properties of polymer membranes modified by carbon nanomaterials as nanofillers differ markedly from the pristine polymer membranes. To widen the applications, attention must be taken to enhance the polymer nanocomposite membrane stability and separation efficiency, specifically in aggressive and adverse environments, by controlling the loading of carbon nanomaterials, interaction between polymer-carbon nanomaterials, their dispersibility, and other minor parameters are still needed to solve the current problems.

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
