# Peer review of "Engineered Zero-Dimensional Fullerene/Carbon Dots-Polymer Based Nanocomposite Membranes for Wastewater Treatment"

_molecules, 2020, doi:10.3390/molecules25214934_

Round 1
Reviewer 1 Report
This is a review on the preparation of composite membranes containing 0D fullerene/carbon dots. As authors mention there are not much work done in the use of 0D carbon forms in the preparation of composite membranes. This review could bring more attention on the research done in this area.
I recommend publication after major revision. I advise the authors to take the following points into account when revising their manuscript;
- Please improve the English. Sometimes it is really difficult to understand what the authors are trying to state.
- Please add a list of abbreviations and a table of content.
- Please add a section about the properties of 0D carbon material and their advantages compared to other fillers used for membranes.
- What is the difference between CQD and GDC in terms of properties/ structure? This could help non-specialist readers.
- Carbonized polymer dots are mentioned but there was no mention of their use in membranes!
- Please add a section on type of membranes prepared using these fillers (e.g. Flat sheet, hollow fibres, etc.)
- Please add a table summarizing the size of the fillers, resulting membrane pore size and filtration ranges.
- Before conclusions please add a section comparing the performance of the membranes prepared with the discussed fillers and other type of fillers. State which are more efficient, performant, cost effective, environmentally friendly, etc.
Author Response
Dear Reviewer,
We are thankful for your comments/suggestions. Your comments/suggestions are enormously helpful to improve the quality of the review article and providing effective inputs for further experimentation in this area for the researchers. Kindly find herewith the answers to all your comments.
All the replies to your questions are attached in the pdf file herewith.
Thank you,
With kind regards,
Mona

Reviewer 2 Report
The article: “Engineered Zero-Dimensional Fullerene/Carbon Dots-Polymer Based Nanocomposite Membranes for Waste Water Treatment” is a very interesting and fits the subject of Molecules. The result analysis is very accurate and adequate. The authors, using the appropriate equipment, have thoroughly investigated the issue.
The following points should be taken into account: 1.It would be good to compare the performance of prepared membranes with these fillers and other types of fillers. A condition that is more efficient, efficient, cost-effective, environmentally friendly, 2.Please add a list of abbreviations and a table of content. 3.If it would be possible please explain is there a difference: between CQD and GDC in terms of properties / structure?
Author Response

(The authors gave the same response as above.)

Reviewer 3 Report
The review work reported in this manuscript (molecules-738937) is interesting and well presented. However, it needs improvements before the acceptance. The work requires major revision.
Comment 1: The plagiarism report of the review article is *35%*, while the standard of the journal is lower than *25%*. Please refer to the plagiarism report attached and reduce the similarity rate.
Comment 2: There are some typographical errors in the manuscript, so authors need to correct it in the revised manuscript.
Comment 3: The authors need to be add the graphical representation figure and explanation of publication trend (1990-2020) in the field of carbon based polymer nanocomposites membranes for wastewater treatment.
Comment 4: Authors need to add the brief introduction of nanotechnology and various nanomaterials (for e.g. carbonaceous nanostructures, nanoparticles and nanocomposites) in the revised manuscript.
Comment 5: In references, the authors mentioned some places with full journal name and some places with abbreviation of the journal (for example. ref no. 15, 25 and 35). Therefore, they need to revise the references according to the journals standard format.

Author Response

(The authors gave the same response as above.)

Round 2
Reviewer 1 Report
The authors have responded to my comments and the manuscript has been revised adequately. I recommend the acceptance of this revised version without further modifications.
Reviewer 3 Report
I believe the revised manuscript has been significantly improved and now warrants publication in Molecules